# Involving the Person with Dementia in Crisis Planning: Focus Groups with Crisis Intervention Teams

**DOI:** 10.3390/ijerph17155412

**Published:** 2020-07-28

**Authors:** Alessandro Bosco, Justine Schneider, Claudio Di Lorito, Emma Broome, Donna Maria Coleston-Shields, Martin Orrell

**Affiliations:** 1Division of Psychiatry and Applied Psychology, University of Nottingham, Nottingham NG7 2TU, UK; Dons.Coleston-shields@nottshc.nhs.uk (D.M.C.-S.); m.orrell@nottingham.ac.uk (M.O.); 2School of Sociology and Social Policy, University of Nottingham, Nottingham NG7 2TU, UK; justine.schneider@nottingham.ac.uk; 3Division of Rehabilitation, Ageing and Wellbeing, University of Nottingham, Nottingham NG7 2TU, UK; claudio.dilorito1@nottingham.ac.uk; 4Hearing Sciences, Division of Clinical Neuroscience, University of Nottingham, Nottingham NG7 2TU, UK; mszeb2@exmail.nottingham.ac.uk

**Keywords:** dementia, co-production, crisis teams, focus group

## Abstract

Dementia leads to progressive critical situations that can escalate to a crisis episode if not adequately managed. A crisis may also resolve spontaneously, or not resolve after receiving professional support. Because of the intensity of the crisis, the extent to which the person engages in decision making for their own care is often decreased. In UK mental health services, ‘crisis teams’ work to avert the breakdown of support arrangements and to avoid admissions to hospital or long-term care where possible. This study aimed to explore the views of crisis teams about promoting the involvement of the person with dementia in decision-making at all points in the care pathway, here defined as co-production. The staff of crisis teams from three NHS Trusts in the UK were interviewed through focus groups. Data were analysed using framework analysis. Three focus groups were run with 22 staff members. Data clustered around strategies used to promote the active involvement of the person with dementia, and the challenges experienced when delivering the care. Staff members reported that achieving a therapeutic relationship was fundamental to successful co-production. Miscommunication and/or lack of proper contact between the team and the individuals and carers receiving support adversely affected the quality of care. Making service users aware of the support provided by crisis teams before they need this may help promote a positive therapeutic relationship and effective care management.

## 1. Introduction

A mental health crisis often means that the person with dementia does not feel able to cope or be in control of a situation [1]. Several factors may be associated with an episode of crisis, such as difficult behaviours (e.g., agitation/aggression, delusions, wandering, dysphoria, resistance to care) and/or physical illness or stress in the carer [2]. When the crisis is not properly managed, it can lead to increased use of emergency care services and hospital or long-term care admission [3].

A given situation may not necessarily be challenging for everyone, and an episode of crisis can be experienced differently by different individuals. For example, a difficult situation may be interpreted as critical by the carer and lead to a call for professional help, without necessarily being experienced as a crisis by the person with dementia. A crisis can resolve spontaneously (e.g., the feeling of loss experienced by a person with dementia may disappear when the carer is home) or remain unresolved even after professional input (e.g., the person may become aggressive towards the carer thinking that s/he is a stranger and this episode is repeated over time with no benefits experienced from either medication or aggression/agitation management plans) [2]. In the UK, multidisciplinary teams have been established as a secondary care service to avert crises. These teams have different names in different places, and there are instances where different terminology is used within the same health service [4]. Consequently, there is confusion amongst people with dementia and carers about the roles and responsibilities of such care teams, which we call here teams managing crises in dementia (TMCDs).

TMCDs consist of health professionals from different disciplines. This diverse make-up ideally guarantees a variety of skills needed to deliver effective care and prevent inappropriate admissions to hospital. For example, an occupational therapist in the team may assess whether the bathroom is accessible for the person with dementia to reduce the risks of possible falls and consequent agitation; whereas a mental health nurse may be looking at whether the person is taking the right dose of medication and check whether its use may be triggering agitation). However, unless the roles are well defined, and responsibilities acknowledged within the team, the intended beneficiaries may feel confused by different professionals conveying different information [5]. A complicating factor is the differing views of what constitutes a good outcome. For example, professionals and carers may have different goals and expectations from the person with dementia [5]. In the pursuit of different outcomes by different agents, the involvement of the person with dementia risks being overlooked [5]. This scenario becomes more challenging when a crisis arises and the family carer becomes primarily responsible for making immediate and important care decisions [6].

Promoting the involvement of people with dementia in decision making in crisis care requires a process of negotiation between health professionals, people with dementia and their primary carers. This process is defined here as co-production, defined by the Social Care Institute for Excellence (SCIE) [7] as ‘a meeting of minds coming together to find a shared solution’. Although in the UK, co-production has been increasingly featured in health care initiatives [8,9,10], very few studies have explored clinicians’ attitudes to co-production in dementia care. In particular, little is known about effective strategies to empower people with dementia accessing TMCDs. Without the appropriate involvement of people with dementia and carers, engagement in the care package is likely to be limited, thus impeding the work of crisis teams and adversely affecting outcomes.

The aim of this study was therefore to explore the experience of involving people with dementia in the delivery of crisis care. We aimed to gather the views of health personnel working in crisis teams, in order to understand how this type of care could be improved.

## 2. Materials and Methods

This was a qualitative study looking into the experience of mental health crisis care in people with dementia, their carers, and staff of TMCDs. The study was funded by the Economic and Social Research Council (UK) and received ethical approval from the Health Research Authority, the East Midlands—Derby Research Committee (reference 18/EM/0023).

Focus groups were held with health professionals from three TMCD services in the North-east and East Midland regions in the UK. The objective was to generate a wide range of views from a variety of professions within the TMCD [11]. One focus group was held with each team in order to reproduce the team dynamic. In addition, the views of people with dementia with respect to the overall study findings were gathered through consultation with members of the Patient and Public Involvement (PPI) group at the University of Nottingham. The views of carers and people with dementia are also reported in a companion paper [12].

To achieve co-production in this study, a person with dementia and a family carer were recruited from the (PPI) group at the School of Medicine at the University of Nottingham and formed an advisory team for the study alongside the lead author (AB). They volunteered to act as co-researchers and advised on the planning (i.e., development of study material such as information sheet and consent forms) and delivery of focus groups (i.e., topic guide, booking the interview room) and checked whether our data interpretation reflected real-life scenarios around co-production in dementia crisis care.

### 2.1. Procedure

The lead author (AB) contacted the service manager of each participating team and provided study information sheets and a poster to recruit health professionals from within their service. Health professionals were eligible to participate if they were working in a multidisciplinary community team managing crises in dementia.

Four weeks before the focus groups, all participants received further information about the study’s aims and objectives. This invited them to reflect on their experience of the management of mental health crises in dementia, and how the active participation of the person with dementia could be promoted when making important health care decisions.

All participants were asked to sign a consent form before taking part in the focus groups. In the consent form, it was stated that all information would be kept confidential unless the participant disclosed anything that could identify them as being at risk of harm to themselves or others. Focus group interviews were audio-recorded and took place in one of the meeting rooms in the National Health Service (NHS) premises where the multidisciplinary teams operate. The time allocated for each focus group was 120 min (30 min for consenting, 80 min for the focus group, and 10 min for debriefing).

### 2.2. Focus Group Topic Guide

AB had the role of moderator and prompted participants with questions to guide the discussion over topics that were relevant to the objectives of the study. The topic guide was tailored around key determinants for co-production in care with the person with dementia. For example, how the staff managed the crisis, if they felt they received the necessary training for the management of the crisis in dementia, what types of crises they managed, barriers and facilitators they experienced, and how to promote the active involvement of people with dementia and their carers in the development of the health care plan.

The number of focus groups held was informed by previous literature on research methodology in qualitative studies [13], according to which, three focus groups are sufficient to capture more than 80% of coded themes.

### 2.3. Framework Analysis

Framework analysis [14], informed the interpretation of the findings. This was done inductively, with categories being generated from the data. The analytical steps of familiarization, thematic framework, indexing, mapping and interpretation were followed. This process allowed for changes to the interpretation of the findings during the analytical process, incorporating data from multiple sources (i.e., from multiple focus groups) and iteratively. The PPI co-researcher with dementia helped check that the data categorisation was not pre-determined and that relatively little control was exerted by the researcher over this process. AB became familiar with the data, by listening to the audiotapes, and formulated preliminary ideas around the focus group transcripts. Based on these ideas, broad themes were identified which were then applied to categorise the data. NVivo^®^ 12 [15] was used to code the data. The final steps of mapping enabled the layout of themes into a tree diagram which portrayed the associations between themes intended to reflect the beliefs and experiences of participants in the study.

To maximise the quality of data analysis and limit research bias for the internal validity of the findings, interrater reliability was calculated between the lead author (AB) and an experienced dementia researcher (EB). AB developed a codebook informed by the guidance provided in the Center for Disease Control and Prevention (CDC) Process for Generating Inter-coder Reliability [16], which was used as guidance for 10% of coding by the two raters (AB, EB). The codebook included explanations on how to code specific pieces of texts in the transcripts and included examples of coded passages to facilitate the process of data analysis.

The calculation of interrater reliability was informed by a previous process plan developed by the authors [12] and measured by means of Kappa coefficient (Cohen’s Kappa) [17] and parameters proposed by Landis and Koch [18]: 0.81–1.00 = almost perfect; 0.61–0.80 = substantial; 0.41–0.60 = moderate; 0.21–0.40 = fair; 0.00–0.20 = slight; < 0.00 = poor. Disagreement between raters was resolved through consensus.

## 3. Results

Three focus groups (i.e., one from each of the three services: S1, S2, S3) were conducted with 22 staff members in total (18 females and 4 males; mean age: 45 years old; range: 24–61). The groups lasted on average 57 min. There was a mix of professional backgrounds among staff from three TMCDs (i.e., one clinical psychologist, three service managers, six community mental health nurses, eight community support workers, three occupational therapists, one of whom was a trainee, and one assistant practitioner). All members of staff worked full time except for the trainee student.

Four themes and five subthemes emerged from the thematic analysis (Table 1): care coalition; strategies to promote involvement in decision making; organisational/structural shortcomings – specifically lack of communication and limited resources; and beliefs around crisis. The interrater reliability revealed a high level of agreement with respect to data coding (*k* = 0.82).

### 3.1. Care Coalition

In order to promote co-production in care, participants reported the need to create equal partnerships with everyone involved in the delivery of care. Barriers were encountered when the collaboration was not promoted (e.g., the carer spoke on behalf of the person with dementia), the role of the carer was not acknowledged (e.g., the carer felt her/his role was threatened by the presence of the staff), or the care environment was challenging and negatively impacted on the way the team could effectively deliver the care (e.g., in the presence of severe cognitive impairment).

***Forming a coalition of care.*** When asked how co-production would occur, staff felt that it would be important to create a bond with the person with dementia and the carer, but that this could require some time to develop, especially during the very first meetings when a discussion around the diagnosis is to be made:

‘Sometimes people find it difficult to describe their condition and only after you establish a bond…Sometimes it takes weeks and weeks before you are able to build up that rapport and make them aware of their condition, so this has an impact on the support we provide (S1-01, Male).

***Barriers to co-production.*** In order to promote the active involvement of the person with dementia, staff reported that most cases required being attentive to the role of the carer, and not divesting them of their caring role. This was of utmost importance for creating a positive and productive care environment:

‘They have been carers for so many years and when we get there they feel we get into their self-identity, their role, and all they can see of us is that we are going there to take their role away. It is very scary for them, so we need to be careful and work very creatively so that they do not have that fear. So, I would say that we kind of adapt to the situation’ (S2-01, Female).

Reaching an agreement with the carer about what was needed for the person with dementia often proved difficult. Having the carer on board seemed to be a key element for any therapeutic plan being designed:

‘I think most of the time family carers and their relatives do not understand why what is happening is happening, …and why maybe support is needed…, and it gets quite difficult’ (S1-01, Female).

When carers felt that their caring role was being threatened by the presence of the team, they would reduce the opportunity of the person with dementia to be involved in care discussion as a way to protect them from potentially untrusted health personnel. This scenario was reported by staff as being a barrier to their work. To contain such resistance, the staff explained from the very beginning their supportive and professional work:

‘Sometimes the carer gets the fear to be cut away, so we make sure to explain that we are not cutting them away. There were instances when the carers did not allow them [the person with dementia] to be involved in any discussions for fear of altering any balance in the care environment’ (S1-01, Male).

Barriers were also linked to the physical and or cognitive disability of the carer. As a community mental health nurse commented, it was often the case that when the carer had a cognitive problem there was little scope for collaborative work, as the quality and extent of communication between them was very limited:

‘Sometimes the carers have cognitive problems themselves on a slightly different side of the spectrum, but that may be quite challenging for us to try and make it work in a collaborative way, as they won’t be able to understand what we are trying to deliver’ (S1-01, Female).

Even when the person with dementia was given an opportunity to engage in decision making, the staff felt that the decision made by the person with dementia was not always in their own best interests. For example, one person acted against the advice offered by the team possibly because of traumatic memories generalised to the present situation:

‘I think the person’s personal journey can be an obstacle sometimes because when we are trying to help with personal care, the person can experience it as some kind of abuse so you are trying to make another person be there as well so but it can bring something very dark to the person and this can be very difficult, very distressing for them and they might not want you to do that and this can stop them from receiving the care really’ (S1-01, Male).

Being attuned with the emotional response at the very moment the care is provided requires empathy from staff, in order to balance the point of view of the person with the demands of care. These are instances that may require more time for staff to explain what the caring tasks they are about to perform entail.

### 3.2. Strategies to Promote Involvement in Decision Making

In order to promote equal partnership, there were strategies that the teams needed to employ. These included caring for the carers experiencing difficulties in dealing with everyday care, promoting an inclusive one-to-one communication with the person with dementia, and adequately supporting proxy decision making when this was most required.

***Caring for the carer.*** Staff members felt that in order to create an environment conducive for collaboration, it was important to attend to the needs of the carer, as they would experience the crisis to a similar extent as the person they were caring for. This viewpoint was reported across the three teams, and as one clinical psychologist explained:

‘… if we see that carers are finding [it] especially difficult living with dementia, we can refer them to carer support groups who help them cope with daily aspects of the condition…we mainly focus on the person with dementia and at times we find it difficult to extend our help to the carers as well. We do refer them to the Alzheimer’s Society or other charities’ (S3-01, Female).

Sometimes staff members found extra time to support the carer by talking with them and exploring if and how to decrease their care burden. The teams reported that this proved particularly helpful for carers who had little or no external support from other family members or friends:

‘When the person with dementia is not verbal, it is good for the carer to have us there to have someone to talk to, because with dementia he lost his wife, maybe friends. We would have a chat with the carer when we are there’ (S2-01, Female).

Helping the carer in need may in turn increase opportunities for more involvement of the person with dementia. When the carer’s needs are not met or problems not addressed, they may be less able to support the person to navigate the care environment (e.g., by explaining to the person with dementia what the team is trying to assess, and enabling them to make a decision on the care plan).

***One to one communication with the person with dementia.*** Establishing one-to-one communication with the person with dementia was reported to be crucial to promote their active involvement in the care process. When this was difficult because the carer insisted on answering all the questions, an occupational therapist explained their strategy to promote their active involvement:

‘I think as a team we are quite good at managing that the person has their voice heard because you know the carer may think they are doing well because, you know, they are helping answer questions and then it is about saying “you know, we would like to talk to him for 10 min” so that we get their views on that and we are quite good at that’ (S1-01, Female).

A community mental health nurse felt that:

‘…all our clinical assessment is about the patient and how the patient can benefit from it. I think they are our focal point, of course also the carers, but they may be going through their own turmoil,…so, I think we would advocate for the individual very strongly because they are our patients’ (S1-01, Female).

***Proxy decision.*** Staff members further explained that in circumstances when an active involvement was not possible because of limited cognitive ability to engage in decision making, it was inevitable for the carer to decide on behalf of the person with dementia through proxy decision making:

‘Yes, I mean, we work quite intensively with the person with dementia to grasp what they need or want. But the carer’s help is useful…’ (S2-01, Female).

### 3.3. Organisational/Structural Shortcomings

In order to effectively co-produce care for a crisis, a coalition of care needed to be formed among different care services. The staff felt that a coalition of care was more difficult to achieve because of factors attributable to the structures and organisations within which they worked; notably a lack of communication between services and limited resources.

Miscommunication or a lack of proper contact between these agents proved challenging for the formation of any care coalition and indirectly led to inadequate levels of partnership and quality of support for the person in crisis. The family doctor (GP, general practitioner) emerged as a pivotal agent in the communication chain:

‘…sometimes the information we get from the referrals from GPs really varies, …sometimes we are given very vague information around things that really mean a lot for the patient and ultimately this has an impact on how we can involve the person in our work as everything gets delayed until we get the right information…’ (S3-01, Female).

When questioned on whether good communication could ease the process for the development of the care plan, a community mental health nurse explained:

‘… when there is good and prompt communication with the GP everything works better, we give them information, they have a clear picture of what is going on and are more ready to get on board with us to develop the care plan’ (S3-03, Female).

Maintaining good communication with primary care services was not only important for partnership in care, but it was also crucial to ensure that the most updated information on care management would be filed for the safety of people with dementia. A community mental health nurse summarised her team’s point of view and commented on how the risk of mishandling medication may become an issue within the community without clear information sharing across services:

‘And then if a medication needs to be altered then we keep the GP on the loop. Because it is an issue with medications in the community. I think that the issue is even more so when there is little communication across services…’ (S3-02, Female).

Difficulties in interaction with social services were associated with ineffective crisis care. The turnover of social workers created confusion for those receiving care, as new people would visit them in their homes. The professionals reported that there was a long waiting list to access social services for respite, and this could have consequences for how effectively the dyad could cope with a crisis.

The staff also reported that services operate with very limited material resources and under time pressure, which could affect crisis care quality:

‘…I just think all the services are under stress in terms of resources and they basically all try to push things to other services because they do not have enough time to manage the cases all by themselves’ (S2-01, Female).

### 3.4. Beliefs Around Crisis

Varying beliefs about crisis influenced the degree of agreement on how to manage the situation. Staff were concerned that there was sometimes a mismatch between what the carer considered to be a crisis and what the TMCD staff assessed as the source of the crisis. This could leave teams with the arduous task of managing different crises at once:

‘We have many people who are [suffering] with severe neglect. So they do not take medication, they forget to eat to drink, so you would think that that is the reason for their crisis. But then the carer insists that the crisis is not having to do with that, or that they really prefer to give priority to a different health outcome, which is ‘calming them down’ and ‘not having them eat more’. But they may be aggressive because they need more food or because they eat too much sugar.’ (S2-01, Female).

The misinterpretation of a crisis trigger is not only a problem in terms of finding the best strategy to manage the situation, but it also represents a risk to the person with dementia who is vulnerable. For example, it may result in basic needs being disregarded. This may be especially problematic for people living in the community who are not aware of the existence of secondary crisis services.

In some cases the GP would refer the person with dementia to the crisis team when the team judged after observation that the episode was not a crisis:

‘…Very often we get a referral from the GP and when we go out we see that they did not need us at all. So yeah the definition of crisis is very different for everyone’ (S2-01, Female).

Receiving a wrong referral could affect the workload of crisis teams because it left less time to promote co-production in crisis care.

## 4. Discussion

This study found that collaborative work was a key element in the delivery of care for a dementia crisis. The teams spoke about the barriers to co-production which they encountered and described the strategies they employed in different situations. It emerged that many strategies were adopted in order to establish trust with the person with dementia and their carer. Trust was seen as a key factor for collaborative development of the health care plan. Inevitably, in advanced dementia, proxy decision making became necessary. In some cases, however, carers made proxy decisions when the person with dementia still had the capacity to be consulted [19,20]. In these circumstances, the teams maintained that an effective strategy to involve the person with dementia in decision making consisted in talking to them on a one-to-one basis, without divesting the carers of their role of primary carer. The teams explained that this could be achieved by clarifying the respective roles of the team and the carer.

Although strategies were consistently employed to maximise collaboration, some factors were beyond the team’s control, including poor institutional communication and inadequate resources. When these factors were present, they lessened the quality of involvement of the person with dementia in designing their own care plan. For example, receiving little information from the GP about the prescription of medication led to delays in the development of a care plan for the individual with dementia. Considering that crisis teams provide their professional input over a short period of time (i.e., usually around 6 weeks), any delay may affect the quality of therapeutic alliance that the staff can form with the person with dementia and the extent to which their preferences and views may be taken into account, especially if these differ from those of the carer.

The progressive cognitive deterioration of dementia may lead the person to experience problems in executing their choices, consequently, the role of crisis teams should be to support the person with dementia to make choices. Their role would be to find strategies to compensate for the declining abilities of the person and in this way to facilitate co-production. In crisis care, this would translate in crisis teams identifying (with the help of the carer) the person’s preferences by showing them possible scenarios (e.g., different types of adjustment for mobility, such as mobility scooter or stick). This may reduce social isolation and increase opportunities for independence, which may prove crucial to promoting a sense of self-worth and self-determination.

### Study Strengths and Limitations

This was the first qualitative study exploring co-production for a mental health crisis in dementia. This study increases understanding of how crisis teams work and how to promote co-production in dementia crisis care. The accuracy in the interpretation of study findings was increased by having multiple raters coding the data and by gathering the views of PPI members with lived experience of dementia. We further felt that our data well represented the views of our participants and that we did not miss out on important information in the analysis as topics were recurrent in each group discussion and no new material ensued from the interviews.

Although we investigated the strategies used by the crisis teams to promote active involvement of the person with dementia in their own care, we did not explore other elements of collaborative work (e.g., personality traits). The issue of professional attitudes on roles within the team only subtly emerged from the interviews with teams and further research could unpack the impact of inter-disciplinary teams, and intra-team relations and attitudes to care according to different professional roles.

In addition, the study did not record how long the staff had worked with dementia crisis cases. This knowledge could help understand differences in approach.

As the people with dementia and carers on the teams’ caseloads were in stressful and difficult situations, they were not involved in the study. Although our findings pointed mainly to the carers’ issues as a potential barrier to co-production, it is to be acknowledged that the cognitive impairment of the person with dementia may greatly reduce their ability to be actively involved in the delivery of care nonetheless. While some preliminary work exploring the experience of crises from the perspective of people with dementia and their carers has been conducted [12], in order to reach a better understanding of care delivery for a crisis, further research with people with dementia accessing TMCD across different sites in the UK should be conducted.

## 5. Conclusions

This study highlighted the importance of promoting co-production in care delivery for a crisis in dementia to deliver tailored and effective care and reduce possible relapses and hospital admission. There is a need to optimise the limited time and resources available to TMCD to ensure co-production in care. For example, more equal power and open communication can be fostered between clinicians and the person with dementia. Health campaigns should be designed at the community level to increase awareness on the work of crisis intervention teams and around crisis prevention and symptoms management. Increased awareness may help the person with dementia and their carers with their involvement in care, when to call for professional help and about ways to self-manage (less critical) situations that do not require access to crisis teams.

It is therefore advisable for crisis teams to reach out to people with dementia and carers (perhaps via primary care providers) soon after a diagnosis of dementia is made and at intervals thereafter. This would allow adequate time to build rapport, to ensure that carers and people with dementia can familiarise with the role and responsibility of the teams and that the team has time to develop a tailored care plan (e.g., episode of crisis, receiving external support, designing advance directives before capacity is compromised by a progressive cognitive impairment).

## Figures and Tables

**Table 1 ijerph-17-05412-t001:** Analytic description of coding.

Theme Coding Assigned to the Theme (N)	Subtheme	Example of Coding per Theme
**Care coalition** (N = 33)	Forming a coalition of care	‘Sometimes people find it difficult to describe their condition and only after you establish a bond…Sometimes it takes weeks and weeks before you are able to build up that rapport and make them aware of their condition, so this has an impact on the support we provide’ (S1-01, Male).
	Barriers to co-production	‘…Yeah and it is very difficult to talk to the carer to do it the right way, because you do not want to say to them ‘oh you are doing this wrong’, as you need them on your side…’ (S2-01, Female).
**Strategies to promote involvement in decision making** (N = 37)	Caring for the carer	‘… she has advanced dementia and she has become very resistant with anyone going there and the carer became so exasperated that we felt they both needed help…’ (S2-02, Female).
	One to one communication with the person with dementia	‘… as a team we are quite good at managing that the person has their voice heard …the carer may think they doing well because you know they are helping answering questions and then it is about saying ‘you know we would like to talk to him for 10 minutes’ so that we get their views on that’ (S1-01, Female).
	Proxy decision	‘…we work quite intensively with the person with dementia to grasp what they need or want. But the carer’s help is useful’ (S2-01, Female).
**Organisational/structural shortcomings** (N = 20)		‘…sometimes the information we get from the referrals from GPs really varies, …sometimes we are given very vague information around things that really mean a lot for the patient and ultimately this has an impact on how we can involve the person in our work as everything gets delayed until we get the right information…’ (S3-01, Female).
**Beliefs around crisis** (N = 11)		‘…I guess a crisis in dementia is when the current situation is so terrible that you cannot make sense of what is happening, because with dementia you know it is going actually get worse, …the carer strain or whatever it is no longer manageable in that situation and crises kick in’ (S1-02, Female).

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
