# Peer review of "Involving the Person with Dementia in Crisis Planning: Focus Groups with Crisis Intervention Teams"

_ijerph, 2020, doi:10.3390/ijerph17155412_

Round 1
Reviewer 1 Report
This study reported about the team managing crisis in dementia (TMCD) in UK. It is very important to build a trust with patients with dementia and their caregiver for caring patients with dementia. This study is interesting because it reported the opinions of staff concretely using frame work analysis. I have some comments, which are as follows:
- I agree with the importance of building a trust with patients with dementia and their caregiver. However, it is often difficult. As authors reported, one-to-one communication with the person with dementia and their caregiver is one of the strategies. Was there another strategy to build a trust?
- Authors categorized “caring for the carer” as “strategies to promote involvement in decision making”. I agree that caring for the carer is important to build a trust with carer. Is caring for the carer associated with decision making of patients with dementia?
- What is the role of TMCD? Please clarify concretely.
- What were the characteristics of patients with dementia in this study? What is the severity of dementia? How many patients did have BPSD?
- How many years did the staff members have experience in care of patients with dementia? Were there differences in contents of comments between low and high experience?
- In conclusions, authors mentioned about COVID-19. However, this study is not associated with COVID-19. Isn’t it unnecessary?
- In p.5, line 158, authors wrote “They ]”. Please delete “]”
- In p.7, line 301, authors wrote “inadequate resource)”. Please delete “)”.
Reviewer 2 Report
I was asked to review this study by Bosco et al on involving a person with dementia in crisis planning. The study used three focus groups with crisis intervention teams. I commend you and your team being able to gather extremely busy health care professionals for focus groups, I acknowledge must have been a very difficult task. I have submitted some minor and major revisions below.
Abstract
The opening sentence of the abstract refers to dementia leading to a progressive cognitive deterioration. The former point is not discussed in the main text. I suggest the Authors to slightly change their opening line or incorporate it in the introduction of the manuscript.
The study aims on line 20 and 21 are not clear “to explore the views of crisis teams about promoting the involvement of the person with dementia in care decisions”. I suggest the author makes it clear if this is before or during crisis or both.
Line 26 “Miscommunication between the team….”, in the results section you refer to “miscommunication or lack of proper contact between these agents”, which I see as two separate things. Perhaps this could be amended to include “lack of proper contact” into the abstract.
Intro
Line 56 “SCIE” please state what this is.
Materials and methods
Line 73 The sentence “One group was run for each….” This isn’t very clear and could be reworded.
Line 77 The sentence “To achieve co-production in this study, a co-researcher with dementia and a family carer were recruited from …” It is unclear if you mean a co-worker has dementia?
You discuss PPI and the formation of an advisory team; can you state exactly what they done. For example, were the research documents co-produced and developed together such as the participant information sheets, consent forms, advertisements etc.
Under procedure - confidentiality is not discussed. A sentence needs to be added to say that this was discussed with participants (providing this was discussed).
The PPI co researcher discussed on line 112 does this person have dementia? Make this clear if so. See comment above about line 77.
Results
It would be interesting to see the range of ages presented here alongside mean age.
Line 158 – Random bracket
Line 182 – “practice agency” this sentence needs to be made clearer. I am unsure what “practice agency” means in this sentence.
Under barriers to co-production you discuss the physical and cognitive disabilities linked to the carer as a barrier to co-production with the crisis team, there is no mention of the physical/cognitive abilities of the person with dementia, particularly in later stages of the dementia journey! If this was not a finding, it should be discussed in the discussion, as this seems an obvious barrier to co-production.
Discussion
You discussed in the methods that the number of focus groups is informed by previous literature, but it would be good to acknowledge if you believed after data analysis you reached data saturation, and why.
Line 301 – random bracket.
Conclusion
Line 329. The person with dementia and their carers should become more knowledgeable about their involvement in care, when to call for professional help and about ways to self-manage (less critical) situations that do not require access to crises teams. I feel this is a sweeping statement and needs to be reworded. Perhaps you could put the sentence that follows this first (about public health campaigns) and discuss how this could educate and provide knowledge to people with dementia and their carers.
Covid-19 receives its first mention here, about fear of contacting crisis teams. This needs integration with and be mentioned in the discussion. It feels very much like an afterthought and this is an important point.
References
Spacing on reference number 9 does not match that of the other references.
Reviewer 3 Report
Thank you for the opportunity to review this paper.
This work will be contributing to a growing area and understanding about the value of co-production in service provision, service design and in research with people living with dementia and family carers of people living with dementia. It is clearly a very important area, engaging in co-production within teams working with people living with dementia, and their families at a time of crisis which would undoubtedly be challenging, as this is a fraught time in people’s lives.
I am structuring my review under the headings used in the paper – with questions or clarifications that I hope are useful for developing the next iteration of the paper.
Introduction
I wondered about the definition used about a mental health crisis ‘A mental health crisis often means that the person with dementia does not feel able to cope or be in control of a situation’ in the introduction – is it the person living with dementia who feels unable to cope or be in control of the situation?…or is it someone else’s perception of the crisis that means a person is referred to the a mental health crisis team? Can this be discussed more in the introduction – as this is crucial in many ways to the paper.
Also the phrasing that ‘a crisis can resolve spontaneously or remain unresolved’ – I think this needs unpacking further. What are the circumstances that lead to spontaneous resolution or what happens when a situation remains unresolved? The parameters reported in the article do not provide an account of this – as the reader has nothing to understand the perimeters against or what the coding book is exactly?
I think this is inter-related with the way a mental health crisis is defined and how it might be experienced differently by different people, which is noted in the article.
I think the aim of the article is clear.
Materials and Methods
How were the co-researchers recruited? Was this one person living with dementia and one family carer?
And I was not quite clear if there was anyone else on the advisory group?
On line 74 ‘One group was run for each team in order to reproduce the team dynamic. In addition, their views were gathered through consultation with members of the Patient and Public Involvement (PPI) group at the University of Nottingham’ – I think it is important to understand what this further consultation was – I am not clear from this what it is or how it contributes to the overall data, subsequent analysis and findings.
I would like more detail of how the inter-coder reliability worked and more critique of its use within the study – for example how were disagreements in the reliability coding were resolved in the full sample? I also was not sure what the codebook and what it incorporated in order that the coding be validated through this method?
Whilst reading the materials and methods, I wondered who had taken part in the focus groups – and then see that this is detailed under results – I think that this may be a matter of reporting style but wondered if there should be a sub-heading for participants under materials and methods ahead of results – detailing professional diversity of the focus groups and the numbers who took part.
Results
I have some questions around the barriers to co-production and the quote used in the table which is different to the ones unpacked within the text. The quote in the table suggests that the professionals within the TMCD feel that there is a ‘right’ and ‘wrong’ way to deliver care. Is there any analysis of the difference in attitudes and understandings of care, and the way that this feeds into developing co-production relationships? Do the professionals within the team reflect on their own professional roles, the inter-relationship between their roles within the team and with the people living with dementia and family carers they are working with?
I also felt that some further explaining of the quotes was required, I am not sure I understood what the staff member means who states that ‘the carer not allowing the person with dementia as part of the discussion for fear of altering the balance means’?
Can there be more analysis of the quotes used and how they reflect the coding.
Here the barriers to co-production all seem to be created by the carer? Is that really the case?
In the section about beliefs about the crisis, it appears to be a difference between the beliefs of the professionals and the family carers that is being described rather than a difference between the family carers and the people living with dementia in the quotes that are used.
Can there be more exploration of the dilemma identified here by staff, around the difference with what carers feel is going on, and the impact on the person living with dementia, with what seems to be described has resulted because of the carer lack of understanding about the impact on mental health with regards hydration and diet.
Discussion
It would be helpful in the discussion to tease out the analysis with regard to how these elements reported impact on the ability to engage in co-production within a crisis. Whilst the discussion does pinpoint the importance of developing trust within these dynamics, and the challenge that the teams have in the timeframes within which they are often working, it would be useful to have more exploration of some of these challenges. Currently it feels like there is something inevitably challenging in developing co-production in these particular moments in the lives of the people that the team are working with – is this the analysis? Clearly these are particularly intense and potentially highly charged times, with ethical implications, and the professionals hold a great deal of power in these moments. This is eluded to through the recognition of trust which is regarded as of great importance in making decision-making work in a more partnership way, but it appears that there are many barriers preventing co-production from happening. It would be helpful for the discussion to go further in its analysis.
Thank you for the opportunity to read this paper and to make suggestions about its development for the next iteration. It is developing understandings in a specific and very important aspect of care around co-production. I hope these pointers are helpful and look forward to the next iteration and the subsequent publication of this work.
Round 2
Reviewer 1 Report
Most of issues suggested by reviewer were revised correctly. Authors mentioned in the limitations that the study did record how long the staff had worked with dementia crisis cases (p. 8, line 353 - 355.). Is it correct? Authors did not report how long the staff had worked with dementia crisis cases.
Reviewer 2 Report
Thank you for the revisions made to this paper.
I have read through this article and feel that this is much clearer to the reader. It is great to see co-production being considered and researched in managing crisis in dementia care. Well done.
Reviewer 3 Report
Thank you for the opportunity to review the revised version of this paper. I thank the authors for the responses to previous comments and I appreciate the additional explanations in the text where required.
I would still note, that in my opinion, the issue of professional attitudes/reflection on roles within the team is subtly reflected in the quotes being used in the results and whilst I appreciate that the researchers state these issues were not reflected in the focus groups and have not been a part of their analysis. I think then it is an area to be added into the paper limitations as it points to possibilities for further research to unpack the impact of inter-disciplinary teams, and intra-team relations and attitudes to care etc. by different professional roles - as reading these quotes it may well have an impact on co-production.
However, other than this I think the paper should absolutely be recommended for publication; and that it is exploring an area of crisis intervention about which little is known, as the authors rightly note, possibly TMCDs are not even known widely to people living with dementia and their families.
